# Enhanced Production of Carboxymethylcellulase by Recombinant *Escherichia coli* Strain from Rice Bran with Shifts in Optimal Conditions of Aeration Rate and Agitation Speed on a Pilot-Scale

**Chung-Il Park [1], Jae-Hong Lee [2], Jianhong Li [3] and Jin-Woo Lee [1,2,4,***

[1]   Department of Applied Bioscience of Graduate School, Dong-A University, Busan 49315, Korea;
     Chungil123@naver.com
[2]   Department of Biotechnology of Graduate School, Dong-A University, Busan 49315, Korea;
     jalmong@nate.com
[3]   College of Plant Science and Technology, Huazhong Agricultural University, Wuhan 430070, China;
     jianhl@mail.hzau.edu.cn
[4]   Department of Biotechnology, College of Life Science and Natural Resources, Dong-A University,
     Busan 49315, Korea
*   Correspondence: jwlee@dau.ac.kr; Tel.: +82-51-200-7593

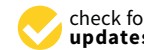

**Featured Application: The simple and economical process with shifts in optimal conditions of the aeration rate and agitation speed of a pilot-scale bioreactor from the cell growth to its production at the mil-log phase was developed for the industrial-scaled mass production of cellulases.**

**Abstract:** The optimal conditions including the aeration rate and agitation speed of bioreactors for the production of carboxymethylcellulase (CMCase) by a recombinant *Escherichia coli* KACC 91335P, expressing CMCase gene of *B. velezensis* A-68, were different from those for its cell growth. The enhanced production of CMCase by *E. coli* KACC 91335P with the conventional multistage process needs at least two bioreactors. Shifts in the optimal conditions of the aeration rate and agitation speed of the bioreactor from the cell growth of *E. coli* KACC 91335P to those for its production of CMCase were investigated for development of the simple and economic process with the high productivity and low cost. The production of CMCase by *E. coli* KACC 91335P with shifts in the optimal conditions of the aeration rate and agitation speed from the cell growth to its production of CMCase in a 100 L pilot-scale bioreactor was 1.36 times higher than that with a fixed optimal conditions of the aeration rate and agitation speed for the production of CMCase and it was even 1.54 times higher than that with a fixed optimal conditions of the aeration rate and agitation speed for cell growth. The best time for the shift in the optimal conditions was found to be the mid-log phase of cell growth. Owing to the mixed-growth-associated production of CMCase by *E. coli* KACC 91335P, shifts in the optimal conditions of the aeration rate and agitation speed of bioreactors from the cell growth to its production of CMCase seemed to result in relatively more cells for the participation in its production of CMCase, which in turn enhanced its production of CMCase. The process with a simple control for shifts in the aeration rate and agitation speed of a bioreactor for the enhanced production of CMCase by *E. coli* KACC 91335P on the pilot-scale can be directly applied to the industrial-scaled production of cellulase.

**Keywords:** aeration rate; agitation speed; carboxymethylcellulase; *Escherichia coli* KACC 91335P; pilot scale; rice bran

## 1. Introduction

Lignocellulosic agricultural byproducts rich in complex carbohydrates can be used as renewable feed-stocks for the production of monomeric sugars for fermentation by physicochemical and enzymatic degradations [1]. For complete enzymatic hydrolysis, at least three types of cellulases are required; endocellulases, exocellulases and cellobiases [2]. Endocellulases (EC 3.2.1.4) break the noncovalent interactions present in the amorphous structure of cellulose, exocellulases (EC 3.2.1.91) hydrolyze chain ends to break the polymer into smaller sugars and cellobiases (EC 3.2.1.21) hydrolyze disaccharides and tetrasaccharides into glucose [3]. The carboymethylcellulase (CMCase) is a type of endocellulase, which is the major cellulase in the conversion of cellulose [2]. However, the high cost and low productivity of cellulases are drawbacks to overcome the enzymatic saccharification of lignocellulosic materials [4]. Many studies on the types of strains including recombinants, substrates and cultural conditions for the enhanced production of cellulases have been reported [5,6].

Most commercial enzymes are produced in large-scale stirred-tank bioreactors [7,8]. The oxygen transfer into a bioreactor is one of the most important factors in upscaling [9]. The oxygen transfer rate (OTR) is affected by the aeration rate and agitation speed as well as the inner pressure of the pilot scale bioreactor [7,10]. The optimal conditions involved in OTR for cell growth are different from those for the production of cellulases [11,12]. Productions of antibiotic by *Xenorhadus nematophila* and lipase by *Bacillus subtilis* were improved with two-stage dissolved oxygen control [13,14]. The production of CMCase by *Escherichia coli* KACC 91334P (formerly *E. coli* JM109/A-53) was enhanced with shifts in the optimal pH and/or temperature from the cell growth to the production of CMCase [15].

The production of the CMCase by *B. velezensis* A-68 under optimized conditions was 83.8 U/mL, whereas that by *E. coli* KACC 91335P was 880.2 U/mL [16]. The production by the recombinant *E. coli* stain was 10.5 times higher than that by the wild of *Bacillus* strain. The CMCase produced by *B. velezensis* A-68 seemed to be essential for its survival, whereas that by *E. coli* KACC 91335P was not essential for its cell growth. The CMCase produced by *B. velezensis* A-68 was a growth-associated product, whereas that by *E. coli* KACC 91335P was a mixed-growth-associated product [16]. The optimal conditions for cell growth of *E. coli* KACC 91335P were different from its production of CMCase. The optimal conditions of the aeration rate and agitation speed of 7 L bioreactors for the cell growth of *E. coli* KACC 91335P were 500 rpm and 0.50 vvm, respectively. However, those for the production of CMCase by *E. coli* KACC 91335P were 416 rpm and 0.95 vvm [16]. For the enhanced production of CMCase, there should be enough cells of *E. coli* KACC 91335P to participate in the production of CMCase. The enhanced production of mixed-growth-associated products with the conventional process needs at least two bioreactors. The effects of shifts in the optimal conditions of the aeration rate and/or agitation speed of bioreactors from the cell growth to its production of CMCase were investigated for the enhanced production of CMCase by *E. coli* KACC 91335P on the pilot-scale in this study.

## 2. Materials and Methods

### 2.1. Medium and Bacterial Strain

The full-length gene coding for the CMCase of *B. velezensis* A-68 was amplified using two specific primers, 5′-AGGAGGAAAAGATCAGATATGAAACGGTCAATC-3′ (Forward) and 5′-TCCAGTATTTCATCCACAACGCAAACCTCC-3′ (Reverse), which were designed based on DNA sequences of previously cloned cellulase genes of *Bacillus* species. The amplified genes were ligated with the T-tail site of pGEM-T Easy Vector System (Promega Co., Madison, WI, USA) and constructed plasmids were transformed into *E. coli* JM109. The constructed recombinant was named *E. coli* Korean Agricultural Culture Collection (KACC) 91335P [17]. This strain was grown at 37 °C in a Luria-Bertani (LB) broth supplemented with 100 µg/mL ampicillin. The carbon and nitrogen sources in the medium for the production of CMCase by *E. coli* KACC 91335P were 132.3 g/L rice bran and 4.68 g/L ammonium chloride, respectively and mineral components in the medium were 5.0 g/L $K_2HPO_4$, 0.6 g/L $MgSO_4 \cdot 7H_2O$, 1.5 g/L NaCl and 0.6 g/L $(NH_4)_2SO_4$ [17].

## 2.2. Production of CMCase by E. coli KACC 91335P

Starter cultures were prepared by transferring cells of *E. coli* KACC 91335P from agar slants to 120 mL of medium in 500 mL Erlenmeyer flasks [16]. They were incubated at 37 °C for 48 h in the shaking incubator for the maintenance of aerobic conditions. Each resulting culture was used for an inoculum to produce CMCase by *E. coli* KACC 91335P. The production of CMCase by *E. coli* KACC 91335P were carried in 7 L bioreactors and 100 L pilot-scale bioreactors (Ko-Biotech Co., Inchen, Korea) [16]. The type of 7 and 100 L bioreactors used in this study was the typical stirred tank fermenter. The type of impellers equipped with bioreactors was the disc-mounted flat-blade turbine. Three six-blade impellers were equipped with each bioreactor. The working volumes of the 7 L and 100 L bioreactors were 3.5 L and 70 L, respectively. The inoculum size and temperature of the batch fermentations were 5% (v/v) and 37 °C. The inner pressure of a 100 L pilot-scale bioreactor for batch fermentations was maintained at 0.02 MPa.

## 2.3. Analytical Methods

The cells in the culture broth were collected by centrifugation at 6000 rpm for 15 min (Fisher Scientific accuSpin 3R Tabletop Centrifuge, Minneapolis, MN, USA). The dry cells weight (DCW, w/v) in the culture broth was determined by weighing the cells after drying at 100–105 °C to a constant weight [18]. The activity of the CMCase was measured in a reaction mixture, which contained 1.0% (w/v) carboxymethycellulose (CMC) and 0.05 mol/L sodium citrate buffer (pH 5.0). The reaction mixture was added into the appropriately diluted cultural broth after removing cells. After incubation at 50 °C for 20 min, the reaction was stopped by the addition of 3,5-dinitrosalicylic acid (DNS) solution. The treated reaction mixtures were boiled for 10 min and cooled in tap water for color stabilization. Their optical densities were measured at 550 nm [19]. A calibration curve was prepared with an authentic glucose (Sigma-Aldrich, Dorset, UK) [1].

The specific growth rate of cell ($\mu_g$) during the log phase of *E. coli* KACC 91335P and the specific production rate ($q_p$) of CMCase were calculated using the following Equations (1) and (2), where X is cell mass concentration (g/L), t is time (h) and P is the concentration of the produced CMCase (U/mL) [20]:

$$\mu_g = (1/X) \times (dX/dt) \tag{1}$$

$$q_p = (1/P) \times (dP/dt) \tag{2}$$

## 2.4. Statistical Analysis

The differences between the experimental data from the effects of shifts in the optimal conditions of the aeration rate and/or agitation speed on cell growth and its production of CMCase were analyzed using Duncan's new multiple-range test. The samples of each run were at least triplicated. The significance level of 0.05 in the statistical analyses corresponds to 95% confidence level in a one-way analysis of variance (ANOVA) [21].

## 3. Results and Discussion

### 3.1. Effects of Shift in Aeration Rate on Production of CMCase

The effects of the shift in the aeration rate of bioreactors on the cell growth of *E. coli* KACC 91335P and its production of CMCase were examined in 7 L bioreactors. The optimal aeration rate of a 7 L bioreactor for the cell growth of *E. coli* KACC 91335P was 0.50 vvm, whereas that for the production of CMCase was 0.95 vvm [17]. The optimal conditions of the aeration rate and agitation speed of the 7 L bioreactor for the cell growth of *E. coli* KACCP 91335P and its production of CMCase were optimized using the response surface methodology based on the cell growth measured as DCW and produced CMCase [17]. Carbon source was 132.3 g/L rice bran and nitrogen source for the production of CMCase was 4.68 g/L ammonium chloride, which had been optimized in a previous study [16]. The aeration

rates of the 7 L bioreactors for experiments were (1) constant at 0.50 vvm, (2) a shift in the aeration rate from 0.50 to 0.95 vvm at 24 h, (3) a shift in the aeration rate from 0.50 to 0.95 vvm at 36 h, (4) a shift in the aeration rate from 0.50 to 0.95 vvm at 48 h and (5) constant at 0.95 vvm. The agitation speed of 7 L bioreactors was fixed at 416 rpm, which was the optimal aeration rate for the production of CMCase by *E. coli* KACC 91335P [16]. As shown in Figure 1, the maximal cell growth was obtained at a constant aeration rate of 0.50 vvm, whereas the highest production of CMCase was obtained with the shift in the aeration rate from 0.50 to 0.95 rpm at 36 h, at which the production of CMCase was 891.4 U/mL.

The cell growth of microorganisms in a batch culture can be divided into four or five different phases—lag phase, log phase, stationary phase, deceleration phase and death phase [22]. Based on the cell growth of *E. coli* KACC 91335P, the times at 24, 36 and 48 h can be considered the early, mid- and late-log phase of the microbial cells growth [23]. The shift in the optimal agitation speed from the cell growth to its production of CMCase at the mid-log phase of cell growth resulted in a relatively higher cell growth of *E. coli* KACC 91335P to participate in the production of CMCase, which in turn enhanced the production of CMCase [24]. The shift in the optimal aeration rate from the cell growth of *E. coli* KACC 91335P to its production of CMCase at the mid-log phase of cell growth also resulted in a relatively higher cell growth of *E. coli* KACC 91335P, which in turn enhanced the production of CMCase. The production of CMCase with a shift in the optimal aeration rate from the cell growth of *E. coli* KACC 91335P to its production of CMCase at the mid-log phase of cell growth was 1.09 times higher than that with a fixed optimal aeration rate for the production of CMCase and even 1.40 times higher than that with a fixed optimal aeration rate for cell growth.

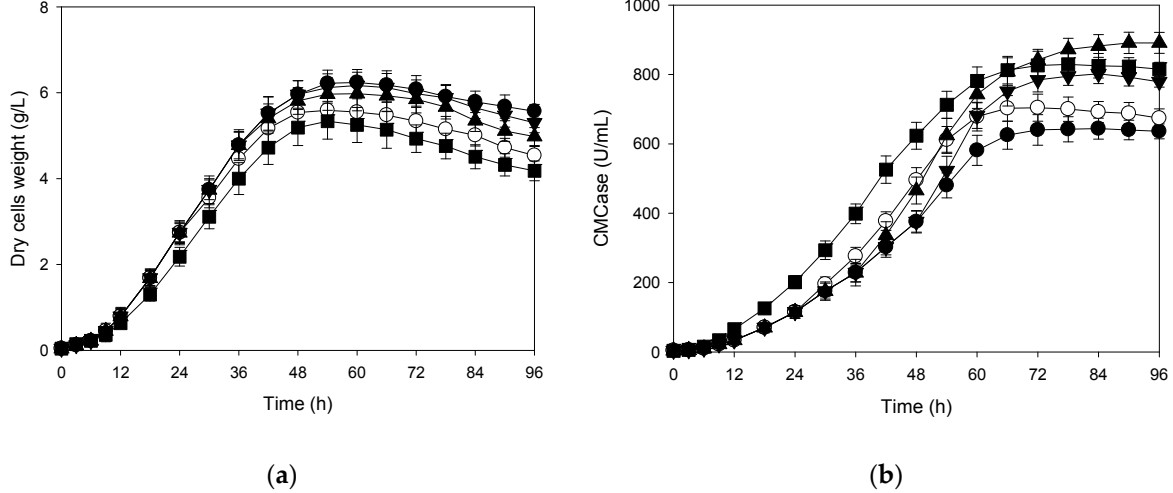

(**a**) (**b**)

**Figure 1.** Effects of shift in aeration rate on (**a**) cell growth and (**b**) production of carboxymethylcellulase by *E. coli* KACC 91335P (●, constant at 0.50 vvm; ■, shift in aeration rate from 0.50 to 0.95 vvm at 24 h; ▲, shift in aeration rate from 0.50 to 0.95 vvm at 36 h; ▼, shift in aeration rate from 0.50 to 0.95 vvm at 48 h; and ○, constant at 0.95 vvm).

*3.2. Effects of Shift in Agitation Speed on Production of CMCase*

The effects of the shift in the agitation speed of bioreactors on the cell growth of *E. coli* KACC 91335P and its production of CMCase were also examined in 7 L bioreactors. The optimal agitation speed of a 7 L bioreactor for the cell growth of *E. coli* KACC 91335P was 500 rpm, whereas that for the production of CMCase was 416 rpm [17]. The agitation speeds of the 7 L bioreactors for experiments were (1) constant at 500 rpm, (2) a shift in the agitation speed from 500 to 416 rpm at 24 h, (3) a shift in the agitation speed from 500 to 416 rpm at 36 h, (4) a shift in the agitation speed from 500 to 416 rpm at 48 h and (5) constant at 416 rpm. The aeration rate of the 7 L bioreactors was fixed at 0.95 vvm, which was the optimal aeration rate for the production of CMCase by *E. coli* KACC 91335P [16]. As shown in Figure 2, the maximal cell growth of *E. coli* KACC 91335P was obtained at a constant

agitation speed of 500 rpm, whereas the highest production of CMCase was obtained with a shift in the agitation speed from 500 to 416 rpm after 36 h, at which the production of CMCase was 874.3 U/mL.

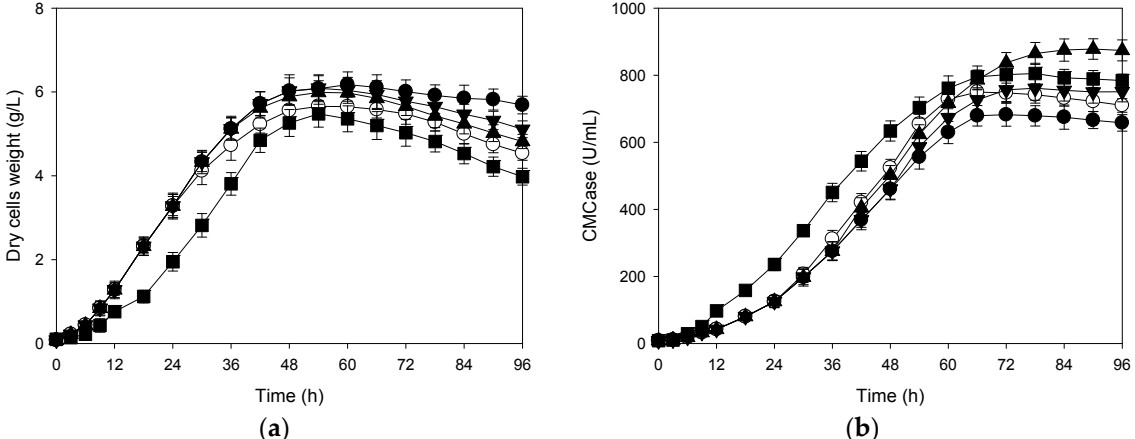

**Figure 2.** Effects of shift in agitation speed on (**a**) cell growth and (**b**) production of CMCase by *E. coli* KACC 91335P (●, constant at 500 rpm; ■, shift in agitation speed from 500 to 416 rpm at 24 h; ▲, shift in agitation speed from 500 to 416 rpm at 36 h; ▼, shift in agitation speed from 500 to 416 rpm at 48 h; and ○, constant at 416 rpm).

### 3.3. Effects of Shifts in Aeration Rate and Agitation Speed on Production of CMCase

The effects of shifts in the aeration rate and agitation speed on the cell growth of *E. coli* KACC 91335P and its production of CMCase were examined in 7 L bioreactors. The aeration rate and agitation speed of the 7 L bioreactors for experiments were (1) constant at 0.50 vvm and 500 rpm, (2) shifted in the aeration rate from 0.50 to 0.95 vvm and the agitation speed from 500 to 416 rpm at 24 h, (3) shifted in the aeration rate from 0.50 to 0.95 vvm and the agitation speed from 500 to 416 rpm at 36 h, (4) shifted in the aeration rate from 0.50 to 0.95 vvm and the agitation speed from 500 to 416 rpm at 48 h and (5) constant at 0.95 vvm and 416 rpm. As shown in Figure 3, the maximal cell growth was obtained at the aeration rate of 0.50 vvm and the agitation speed of 500 rpm, which were known as the optimal conditions for the cell growth of *E. coli* KACC 91335P. However, the highest production of CMCase was obtained with shifts in the aeration rate from 0.50 to 0.95 vvm and the agitation speed from 500 to 416 rpm at 36 h, the mid-log phase of cell growth, which the production of CMCase was 953.2 U/mL. The production of CMCase with shifts in the optimal conditions of the agitation speed and aeration rate from the cell growth of *E. coli* KACC 91335P to its production of CMCase at the mid-log phase of cell growth was 1.21 times higher than that with the fixed optimal agitation speed and aeration rate for the production of CMCase. As shown in Table 1, the cell growth of *E. coli* KACC 91335P with shifts in the optimal conditions of the aeration rate and agitation speed at 36 h was also 1.21 times higher than that with the optimal conditions of the aeration rate and agitation speed for the production of CMCase.

The production pattern of microbial metabolites can be divided in growth-associated, mixed-growth-associated and non-growth-associated [20]. The formation of mixed-growth-associated products takes place during the log and early stationary phase. In this case, the specific production rate ($q_p$) with the specific growth rate ($\mu_g$) is given by the following Equation (3):

$$q_p = \alpha\mu_g + \beta \tag{3}$$

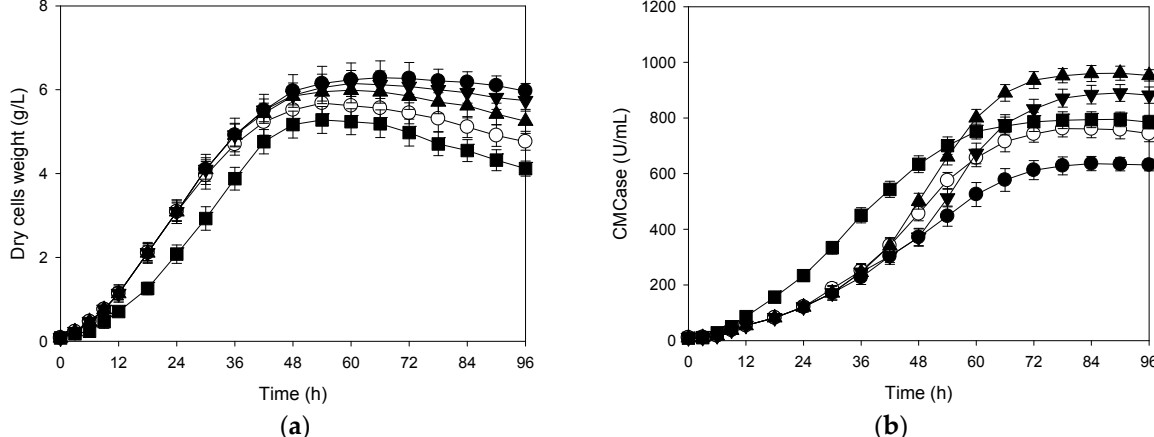

**Figure 3.** Effects of shift in aeration rate and agitation speed on (a) cell growth and (b) production of CMCase by *E. coli* KACC 91335P (●, constant at 0.50 vvm and 500 rpm; ■, shifts in aeration rate from 0.50 to 0.95 vvm and agitation speed from 500 to 416 rpm at 24 h; ▲, shifts in aeration rate from 0.50 to 0.95 vvm and agitation speed from 500 to 416 rpm at 36 h; ▼, shifts and aeration rate from 0.50 to 0.95 vvm and in agitation speed from 500 to 416 rpm at 48; and ○, constant at 0.95 vvm and 416 rpm).

**Table 1.** Effects of shifts in aeration rate and/or agitation speed on cell growth and production of CMCase by *E. coli* KACC 91335P for 96 h.

| Condition | Aeration (vvm) | Agitation (rpm) | Time for Shift (h) | Dry Cells Weight (g/L) | CMCase (U/mL) |
|---|---|---|---|---|---|
| Shift in agitation | 0.95 | 500 | - | 5.69 ± 0.21 [ae1,2] | 658 ± 25 [ad] |
| | 0.95 | 500 to 416 | 24 | 4.54 ± 0.17 [bf] | 710 ± 20 [ag] |
| | 0.95 | 500 to 416 | 36 | 4.82 ± 0.17 [bc] | 874 ± 31 [b] |
| | 0.95 | 500 to 416 | 48 | 5.11 ± 0.20 [cg] | 751 ± 30 [cg] |
| | 0.95 | 416 | - | 3.98 ± 0.20 [d] | 784 ± 20 [ce] |
| Shift in aeration | 0.50 | 416 | - | 5.67 ± 0.16 [ae] | 636 ± 22 [d] |
| | 0.50 to 0.95 | 416 | 24 | 4.54 ± 0.21 [bf] | 674 ± 27 [ad] |
| | 0.50 to 0.95 | 416 | 36 | 4.95 ± 0.15 [bcg] | 891 ± 30 [b] |
| | 0.50 to 0.95 | 416 | 48 | 5.30 ± 0.23 [ag] | 779 ± 16 [ce] |
| | 0.95 | 416 | - | 4.18 ± 0.23 [df] | 815 ± 20.1 [e] |
| Shifts in agitation and aeration | 0.50 | 500 | - | 5.97 ± 0.18 [e] | 631 ± 21 [d] |
| | 0.50 to 0.95 | 500 to 416 | 24 | 4.76 ± 0.20 [bc] | 744 ± 29 [cg] |
| | 0.50 to 0.95 | 500 to 416 | 36 | 5.25 ± 0.24 [ag] | 953 ± 21 [cf] |
| | 0.50 to 0.95 | 500 to 416 | 48 | 5.74 ± 0.21 [ae] | 882 ± 29 [b] |
| | 0.95 | 416 | - | 4.12 ± 0.18 [df] | 785 ± 25 [ce] |

[1] Values are mean ± SE, n = 3; [2] Values with different letters mean that they are significantly different at $p < 0.05$.

The production of CMCase by *E. coli* KACC 91335P begins with cell growth and continues to the early stationary phase. This is the typical pattern for the mixed-growth-associated production. The shifts in the optimal conditions from the cell growth to those for its production of CMCase after the mid-log phase seemed to result in relatively higher cell growth, which sequentially enhanced the production of CMCase by *E. coli* KACC 91335P. The CMCase produced by *B. velezensis* A-53 was a growth-associated product, whereas that by its recombinant, *E. coli* KACC 91335P, was a mixed-growth associated product [25]. Owing to the difference between the optimal conditions for the cell growth of *E. coli* KACC 91335P and its production of CMCase, higher cell growth under optimal conditions for cell growth resulted in a lower production of CMCase. However, the production of CMCase by *E. coli* KACC 91335P with shifts in the optimal conditions of pH and/or temperature from cell growth to its production of CMCase was 1.27 times higher than that with the fixed optimal conditions of pH and/or temperature for the production of CMCase [15].

### 3.4. Correlation Between Cells Growth and Its Production of CMCase

The correlation between the dry cells weights during the cell growth of *E. coli* KACC 91335P and its production of CMCase was analyzed using SigmaPlot (Systat Software Inc., San Jose, CA, USA) for Windows Version 10.0. A quadratic regression analysis of data obtained from experiments without shifts in optimal conditions of the aeration rate and/or agitation speed gave the second-order polynomial Equation (4), which represented the correlation between the cell growth (X) and its production of CMCase (Y) as shown in Figure 4a.

$$Y = 1132.9 - 77.8X - 1.2X^2 \qquad (4)$$

The multiple correlation coefficient ($R^2$) of above regression equation was 0.9751, which can explain the 97.51% variation in the response. Its value of the adjusted coefficient of determination (Adj. $R^2$ = 0.9442) was relatively high to advocate for this model's significance [26]. This model with the multiple correlation and adjusted coefficients indicated that the above equation was adequate for predicting the production of CMCase by *E. coli* KACC 91335P without shifts in optimal conditions of the agitation speed and/or aeration rate. Based on the regression equation of these experimental data, the optimal conditions of the aeration rate and agitation speed for the cell growth of *E. coli* KACC 91335P were proven to be different from those for its production of CMCase. A higher cell growth of *E. coli* KACC 91335P resulted in a lower production of CMCase.

However, the quadratic regression analyses of the correlation between the cell growth and its production of CMCase with shifts in optimal conditions of the aeration rate and/or agitation speed were different from those without shifts in optimal conditions of the aeration rate and/or agitation speed. The higher cell growth with shifts in the aeration rate and/or agitation speed resulted in the enhanced production of CMCase by *E. coli* KACC 91335P, as shown in Figure 4b–d. The multiple correlation coefficients ($R^2$) obtained from the experimental data with shifts in optimal conditions of the aeration rate and/or agitation speed at 24, 36 and 48 h were 0.7753, 0.9376 and 0.8820, respectively. The second-order polynomial Equation (5) from the correlation between the cell growth (X) and its production of CMCase (Y) with shifts in optimal conditions of the aeration rate and/or agitation speed at 36 h was as follows:

$$Y = -2780.4 + 1305.9X - 113.4X^2 \qquad (5)$$

The CMCase produced by a wild type, *B. velezensis* A-68, was a growth-associated product, whereas that by its recombinant, *E. coli* KACC 91335P, was a mixed-growth-associated product [7,16]. The production of CMCase by *E. coli* KACC 91335P was partially correlated with its cell growth. Owing to the mixed-growth-associated production of CMCase by *E. coli* KACC 91335P, shifts in the optimal conditions of the aeration rate and/or agitation speed from the cell growth to its production of CMCase at the mid-log phase of cell growth resulted in relatively higher cell growth, which in turn enhanced the production of CMCase.

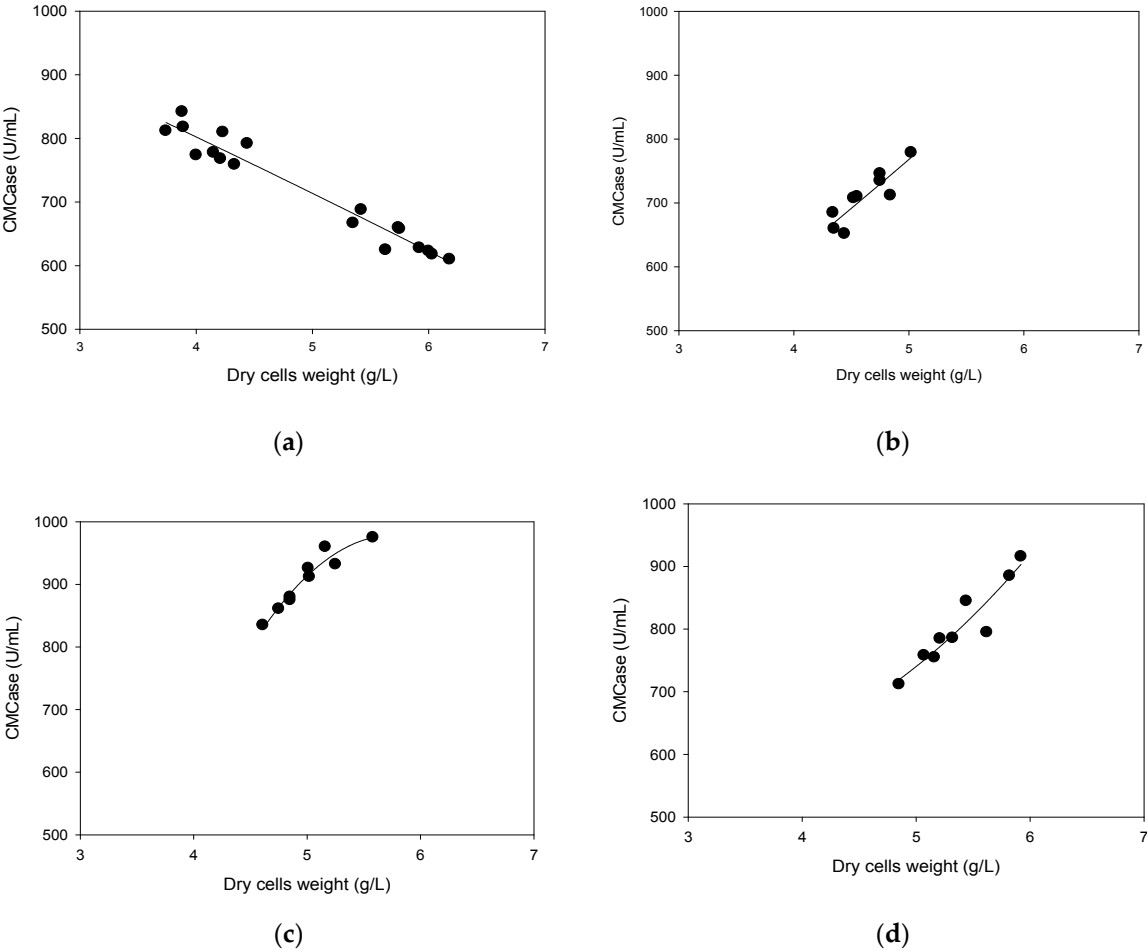

**Figure 4.** Relationship between cell growth and production of CMCase by *E. coli* KACC 91335P (**a**) without shifts in agitation and/or aeration, (**b**) with shifts in agitation and/or aeration at 24 h, (**c**) with shifts in agitation and/or aeration at 26 h and (**d**) with shifts in agitation and/or aeration at 48 h.

*3.5. Production of CMCase with Shifts in Aeration Rate and Agitation Speed of Pilot-Scale Bioreactor*

The optimal agitation speed of a 100 L pilot-scale bioreactor was calculated based on the optimal agitation speed of the 7 L bioreactor and the difference between diameters of impellers in 7 and 100 L bioreactors. The diameters of impellers with the same type in 7 and 100 L bioreactor were 6.0 and 16.6 cm. The impeller tip speed was an essential parameter when the agitating liquid contained solid particles [27]. The impeller tip speed at the agitation speeds of 416 and 500 rpm of a 7 L bioreactor were calculated to be 130.7 and 150.1 cm/s, respectively, which were almost the same as those at 150 and 180 rpm of a 100 L bioreactor. The optimal conditions of the aeration rates and agitation speeds of a 100 L bioreactor for the cell growth of *E. coli* KACC 91335P were decided to be 0.50 vvm and 180 rpm, respectively, whereas those for its production of CMCase were 0.95 vvm and 150 rpm. As shown in Figure 5a,b, the cell growth of *E. coli* KACC 91335P and its production of CMCase for 96 h with a fixed optimal conditions of the aeration rate and agitation speed were 5.58 g/L and 784.2 U/mL, respectively, whereas those with a fixed optimal conditions of the aeration rate and agitation speed for the production of CMCase were 3.45 g/L and 893.1 U/mL. However, the cell growth of *E. coli* KACC 91335P and its production of CMCase with shifts in the optimal condition of the aeration rate and agitation speed from the cell growth to its production of CMCase at 36 h were 5.27 g/L and 1216.5 U/mL, respectively, as shown in Figure 5c.

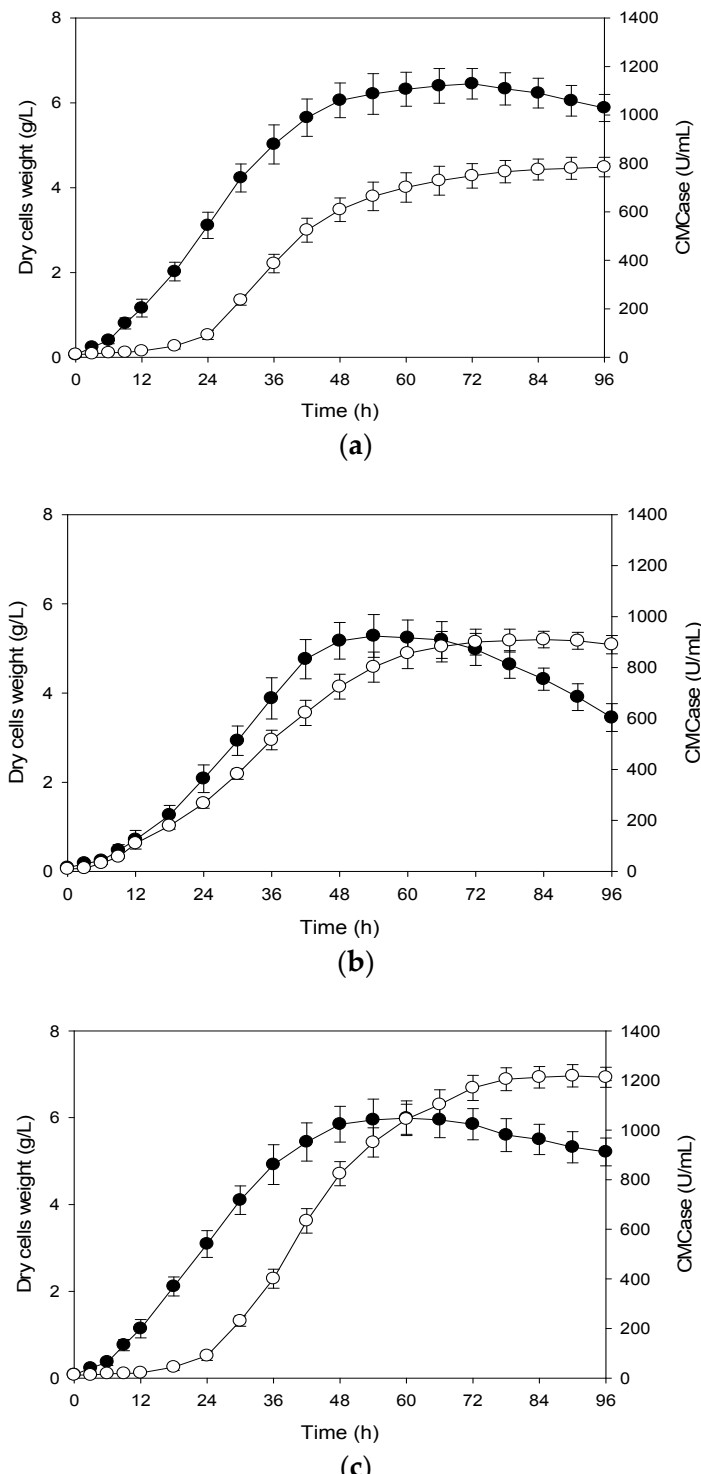

**Figure 5.** Comparison of cell growth and production of CMCase by *E. coli* KACC 91335P (**a**) under fixed optimal conditions of aeration rate and agitation speed for cell growth, (**b**) those under fixed optimal conditions of aeration rate and agitation speed for production of CMCase and (**c**) with shifts in optimal conditions of aeration rate and agitation speed from cell growth to production of CMCase at 36 h; ●, dry cells weight and ○, CMCase.

As shown in Table 2, the growth rate constant ($\mu$) values of a the fixed optimal conditions of the aeration rate and agitation speed for the cell growth and those for the production of CMCase were 0.059 and 0.044/h, respectively, whereas that with shifts in the optimal conditions of the aeration rate

and agitation speed from the cell growth to its production of CMCase at 36 h, the mid-log phase of cell growth, was 0.052/h. However, $\mu_{max}$ with shifts in the optimal conditions of the aeration rate and agitation speed during a specific time was higher than those without shifts. As shown in Figure 6, the shifts in the optimal conditions of the aeration rate and agitation speed of the 100 L pilot-scale bioreactor from the cell growth to its production of CMCase at the mid-log phase resulted in a relatively more cells to participate in their production of CMCase, which in turn enhanced the production of CMCase. The production of CMCase by *E. coli* KACC 91335P in a 100 L bioreactor with shifts in the optimal conditions of the aeration rate and agitation speed from the cell growth to its production of CMCase at the mid-log phase was 1.36 times higher than that with a fixed optimal condition of the aeration rate and agitation speed for the production of CMCase and even 1.54 times higher than that with those for the cell growth.

**Table 2.** Comparison of cell growth and production of CMCase by *E. coli* KACC 91335P in a 100 L bioreactor.

| Conditions and Results | | Without Shift | | With Shifts at 36 h |
|---|---|---|---|---|
| Conditions | Aeration (rpm) | 180 | 150 | 180 to 150 rpm |
| | Agitation (vvm) | 0.50 | 0.95 | 0.50 to 0.95 vvm |
| Results | DCW (g/L) | 5.88 | 3.45 | 5.21 |
| | Cellulase (U/mL) | 785 | 890 | 1212 |
| | $\mu$ (/h) | 0.059 | 0.044 | 0.052 |
| | $\mu_{max}$ (/h) | 0.092 | 0.083 | 0.103 |
| | dx/dt (g/L·h) | 0.061 | 0.035 | 0.054 |
| | dp/dt (U/mL·h) | 8.177 | 9.271 | 12.625 |
| | dp/dx (/U/g) | 133.50 | 257.97 | 232.63 |

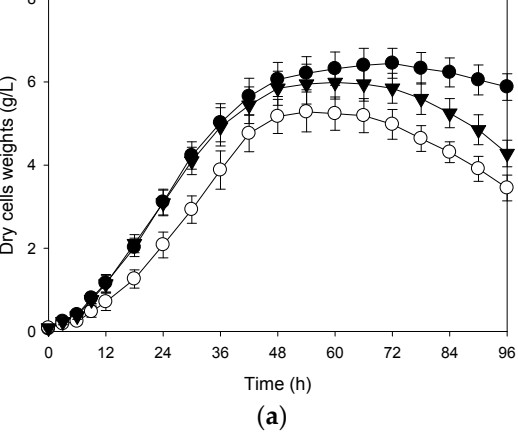
(a)

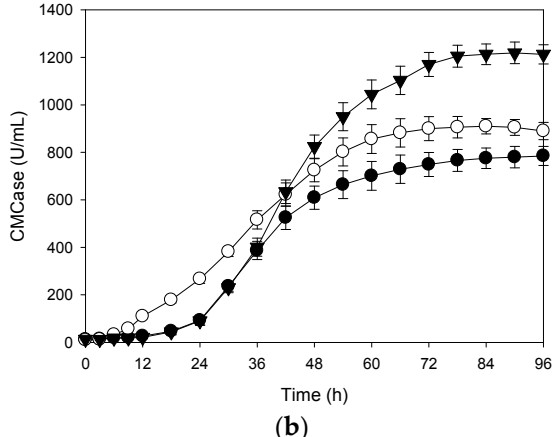
(b)

**Figure 6.** Comparison of (**a**) cell growth and (**b**) production of CMCase by *E. coli* KACC 91335P under the fixed optimal conditions of aeration rate and agitation speed for cell growth (●), under fixed optimal conditions of aeration rate and agitation speed for production of CMCase (○) and with shifts in optimal conditions of aeration rate and agitation speed from cell growth to production of CMCase at mid-log phase (▼).

During fermentation with the multistage bioreactors, the first bioreactor containing microorganisms in the growth phase was for the cell growth and the second bioreactor containing non-proliferating cells was for the production of metabolites [28]. The production of the arachidonic acid by *M. alpina* was increased by using continuous multistage fermentation with precise control of the agitation speed and aeration rate at proper times [29]. The multistage chemostat system may be beneficial for improving the genetic stability and production of metabolites [20,30]. The advantages of the continuous fermentation with optimized process conditions are higher volumetric and long-term

continuous productivity as well as reduced labor costs [31]. As a strategy of multistage continuous cultures for the higher production of CMCase, the first stage should involve the cell growth of *E. coli* KACC 91335P and the second stage should involve the production of CMCase. However, the set-up and maintenance of continuous fermentations, including continuous multistage systems, are more difficult than those for batch fermentations. Significant volumes of medium, including products, may be lost when contamination occurs during continuous multistage fermentation [32]. Moreover, the cost for constructing continuous fermentation equipment with multistage bioreactors is more expensive than that for the batch fermentation. The CMCase produced by *E. coli* KACC 91335P was the mixed-growth-associated product. Without the multistage continuous system, the production of CMCase by *E. coli* KACC 91335P in the batch system of a 100 L pilot-scale bioreactor was enhanced with shits in optimal conditions of the aeration rate and agitation speed at the mid-log phase.

## 4. Conclusions

In this study, the production of CMCase by *E. coli* KACC 91335P in a pilot-scale bioreactor was enhanced by shifting the optimal conditions of the aeration rate and agitation speed from the cell growth to its production of CMCase. The best time for shifts in the optimal conditions of the aeration rate and agitation speed was found to be 36 h, which was the mid-log phase of *E. coli* KACC 91335P. Shifts in the optimal conditions of the aeration rate and agitation speed of the bioreactor may comprise a simple and economical process to enhance the production of CMCase. Shifts in the optimal conditions of the aeration rate and agitation speed from the cell growth to its production of CMCase resulted in a relatively more cells for participating in its production, which in turn enhanced the production of CMCase. The process developed in this study needs only a bioreactor for the enhanced production of CMCase. In addition, this process is easy to control the aeration rate and agitation speed in a bioreactor and does not even need to transfer cells from a bioreactor to the other bioreactor. Rice bran was used as a substrate in the production of CMCase by *E. coli* KACC 91335P, thus the process developed in this study seemed to overcome a major constraint in the enzymatic hydrolysis of lignocellulosic agricultural-byproducts in the preparation of substrates for the production of bioethanol. The simple and economical process developed on the pilot-scale in this study can be applied for the industrial-scaled productions of mixed-growth- associated products.

**Author Contributions:** C.-I.P. and J.-H.L. performed laboratory experiments, analysis, interpretation of data and original draft preparation; J.L. provided technical assistance and edited the manuscript; J.-W.L. provided primary supervision and helped in analysis, methodical guidance, critical reviewing and approval of final version to be submitted.

**Funding:** This study in the article was financially supported by the Dong-A University Research Fund. The authors gratefully acknowledge this support.

**Acknowledgments:** Authors would like to thank Editage (www.editage.co.kr) for English language editing.

**Conflicts of Interest:** The authors declare no conflicts of interest.

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
