# Peer review of "Enhanced Production of Carboxymethylcellulase by Recombinant Escherichia coli Strain from Rice Bran with Shifts in Optimal Conditions of Aeration Rate and Agitation Speed on a Pilot-Scale"

_applsci, doi:10.3390/app9194083_

Round 1

Reviewer 1 Report

The novelty and the significance of the work are unclear given that similar works (references 11 and 12) have already been reported before. It is desirable if further discussion on the current work's importance is added in the manuscript, specifically highlighting how different it is from the previous works and what new information was obtained that were not already reported from the earlier studies. 

The results discussed in the text are not consistent with the data presented on the figures. Specifically, in all cases, they highlight that 36h was the best condition in the text. However, this is not reflected on the graphs. Equations are also not defined, as well as certain abbreviations.

Minor typo comments:

L38: Typo Endocellualses

L42: Odd statement: "major cellulase in the conversion of cellulase"

L54: Was the gene isolated or amplified?

L 62: "was" to were

L70: "And mineral.." should not be a separate statement

L84: Wrong unit for the weight (currently degree celsius is indicated)

L86: What is the actual pH? There is a period after 5

L87: for 20 min? instead of in?

L91 and L170: Equation not defined

Author Response

Comments and Suggestions for Authors

The novelty and the significance of the work are unclear given that similar works (references 11 and 12) have already been reported before. It is desirable if further discussion on the current work's importance is added in the manuscript, specifically highlighting how different it is from the previous works and what new information was obtained that were not already reported from the earlier studies. 

(Response)

The reference 11 and 12 were on the optimization of the agitation speed and aeration rate of a 7 L bioreactor and the inner pressure of a 100 L pilot-scale bioreactor for the production of carboxymethylcellulase (CMCase) and cellobiase, respectively, by wild strain, C. lytica LHB-14. As mentioned in this manuscript, productions of cellulases by wild strains are growth-associated, which means that the optimal conditions for the productions of the cellulases by wild strains are the same as those for cell growth. Those by recombinants are mixed-growth-associated, which means that the optimal conditions for the productions of the cellulases by recombinants are different from those for their cell growth. This study is on the enhanced production of cellulase with shifts in optimal conditions from cell growth to production of cellulase. As results from experiment and analysis of data, we concluded the best time for the shifts was the mid-log phase of cell growth. We described those in the revised manuscript. We marked the revised parts with the red color.

The results discussed in the text are not consistent with the data presented on the figures. Specifically, in all cases, they highlight that 36 h was the best condition in the text. However, this is not reflected on the graphs. Equations are also not defined, as well as certain abbreviations.

(Response)

We found big mistakes in the symbols of figure 1, 2, and 3 in the previous manuscript. We revised the symbols in figures in the revised manuscript. We also added definitions of equations of 1 and 2.

Minor typo comments:

L38: Typo Endocellualses

(Response)

We revised the word as the reviewer’s comment.

L42: Odd statement: "major cellulase in the conversion of cellulase"

(Response)

We revised the word as the reviewer’s comment.

L54: Was the gene isolated or amplified?

(Response)

The gene was amplified using two primers and the amplified genes were ligated with the vector, which were transformed to E. coli JM109. We added the process for the construction of the recombinant, E. coli KCCC 91335P in the revised manuscript.

L 62: "was" to were

(Response)

We revised the word as the reviewer’s comment.

L70: "And mineral.." should not be a separate statement

(Response)

We revised sentences as the reviewer’s comment.

L84: Wrong unit for the weight (currently degree celsius is indicated)

(Response)

We revised the sentence as the reviewer’s comment.

L86: What is the actual pH? There is a period after 5

(Response)

The pH 5.0 is the optimal pH for the hydrolytic activity of the cellulase produced by E. coli KACC 91335P. We revised the sentence as the reviewer’s comment.

L87: for 20 min? instead of in?

(Response)

We revised the word as the reviewer’s comment.

L91 and L170: Equation not defined

(Response)

We added definitions of equations in the revised manuscript as the reviewer’s comment.

Reviewer 2 Report

This article describes the effects of shift in agitation speed and aeration rate on the production of cellulase. I feel that the shift of the two factors was effective, and the enhance of enzyme productivity was significant if the enzyme has enough value to be produced at large scale. However, there are fetal mistakes in this manuscript. At Fig. 1, 2, and 3, open triangle symbols are shown in graphs, but there is no description in the legends. Moreover, the authors mentioned that the shift at 36 h was most effective in the text, but the shift at 48 h showed the highest production. Such contradiction is so confusing for me that I would like to review again after the manuscript is revised correctly.

Author Response

Comments and Suggestions for Authors

This article describes the effects of shift in agitation speed and aeration rate on the production of cellulase. I feel that the shift of the two factors was effective, and the enhance of enzyme productivity was significant if the enzyme has enough value to be produced at large scale. However, there are fetal mistakes in this manuscript. At Fig. 1, 2, and 3, open triangle symbols are shown in graphs, but there is no description in the legends. Moreover, the authors mentioned that the shift at 36 h was most effective in the text, but the shift at 48 h showed the highest production. Such contradiction is so confusing for me that I would like to review again after the manuscript is revised correctly.

(response)

Thank for the reviewer’s very valuable comments for the improved manuscript. We found big mistakes in the symbols of the figure 1, 2, and 3 in the previous manuscript. We revised the symbols in figures in the revised manuscript as the reviewer’s comment. We marked the revised parts with the blue color.

Reviewer 3 Report

Overall and major comments:

This study is looking at different conditions with focus on the agitation speed and aeration rate to enhance production of carboxymethylcellulase when produced by a specific modified strain of E. coli. Overall, this manuscript needs major improvements. The abstract needs re-writing as it is lacking structure with no background, methodology or aims included, but only results. The introduction is superficial and more background is needed in order to understand the overall purpose of this study and the wider impact which in this current form is unclear. There are no aims or objectives. The methodology is missing a lot of detail which makes the reproducibility of this study impossible. The results and discussion section contains multiple paragraphs that should be included in methodology. The discussion is lacking links to literature or justifications. No evidence of critical judgement. This section just feels like an enumeration of findings without any analysis or interpretation.

Specific comments:

Line 28: ‘the simple and economic process’ – This statement needs justification why it is simple and why it is economic.

Line 42: ‘…in the conversion of cellulase’ – In this statement, cellulose would make more sense as cellulose is converted by cellulose.

Line 52: ‘CMCase’ – acronym needs to be explained when used for the first time in text.

Lines 54-61: ‘The gene coding….0.95 vvm [17].’ – This paragraph should be in the Methods section and not in the Introduction.

Lines 60 and 61: Use past tense instead of present tense as the work has already been carried out.

Lines 66-67: ‘The gene coding…E. coli…’ – Repetition from above.

Line 68: ‘LB medium’ – acronym used for the first time in text – needs explaining.

Line 74: ‘120 ml of medium in 5000 mL Erlenmeyer’ – Is this correct? Why would you use such a small volume in such a big flask?

Line 82: ‘12000 xg’ – That seems very high. Is it correct?

Line 84: Replace ‘weight of 100-105oC’ with ‘at’

Line 87: ’20 in’ – Correct with 20 min.

Line 87: ‘DNS method’ – This method should be briefly explained even if referenced.

Line 88: ‘calibration curve prepared..’ – Add ‘was prepared’.

No mention of how many replicates were done for these experiments.

The aeration rates and agitation speeds used were described in the Results section instead of Methods.

No mention of impeller type in either the 7L bioreactor or the 100L bioreactor. In fact no mention of what type of bioreactors are.

No mention of feeding regimes or bioreactor operation modes.

Line 91: Those two equations should be presented independently and not as one.

Line 102: How was the optimal agitation determined and what does optimal mean in this context? It is not clear.

The equations presented in the Results section should be presented in the Methods instead.

Line 229: Past tense instead of present tense.

Agitation speeds presented as rpm are not appropriate when comparing bioreactors at different scales. Instead energy dissipation rates should be used.

Figure 5 contains multiple parameters in one graph which makes it confusing.

Line 263: ‘…for the production of wine’ – It is not clear what the connection to wine production is.

Author Response

Comments and Suggestions for Authors

Overall and major comments:

This study is looking at different conditions with focus on the agitation speed and aeration rate to enhance production of carboxymethylcellulase when produced by a specific modified strain of E. coli. Overall, this manuscript needs major improvements. The abstract needs re-writing as it is lacking structure with no background, methodology or aims included, but only results. The introduction is superficial and more background is needed in order to understand the overall purpose of this study and the wider impact which in this current form is unclear. There are no aims or objectives. The methodology is missing a lot of detail which makes the reproducibility of this study impossible. The results and discussion section contains multiple paragraphs that should be included in methodology. The discussion is lacking links to literature or justifications. No evidence of critical judgement. This section just feels like an enumeration of findings without any analysis or interpretation.

(Response)

Thank for the reviewer’s very valuable comments for the improved manuscript. We described the back ground as well as aims in the Abstract of the revised manuscript. However, there was some limitation of the length in the abstract of this journal. We also described aims and objectives in the Introduction and more detailed methodology in the Materials and Methods in the revised manuscript. We described the significance and originality of this study in the Conclusion in the revised manuscript. We marked the revised parts with the dark green color. However, some parts were overlapped with other reviewers’ comments.

Specific comments:

Line 28: ‘the simple and economic process’ – This statement needs justification why it is simple and why it is economic.

(Response)

As we described in the Results and Conclusion, the CMCase produced by E. coli KACC 91335P was the mixed-growth-associated product. The first stage should involve the cell growth of E. coli KACC 91335P and the second stage should involve the production of CMCase. However, the set-up and maintenance of the multistage fermentation as well as the cost for multistage bioreactors are difficult and expensive. Without the multistage continuous system, the production of CMCase by E. coli KACC 91335P was enhanced with shits in optimal conditions of the aeration rate and agitation speed at the mid-log phase on the pilot-scale bioreactor.. The process developed in this study with shifts in the agitation speed and aeration rate of a bioreactor is simple and economic, which can be directly applied to the mass production of cellulase.

Line 42: ‘…in the conversion of cellulase’ – In this statement, cellulose would make more sense as cellulose is converted by cellulose.

(Response)

The other reviewers also gave us similar comments on this sentence. We revised the sentence as reviewers’ comments.

Line 52: ‘CMCase’ – acronym needs to be explained when used for the first time in text.

(Response)

We revised names of microorganisms as the reviewer’ comments.

Lines 54-61: ‘The gene coding….0.95 vvm [17].’ – This paragraph should be in the Methods section and not in the Introduction.

(Response)

We moved this paragraph into the Materials and Method and described more detailed methods as reviewer’s comments.

Lines 60 and 61: Use past tense instead of present tense as the work has already been carried out.

(Response)

We revised tenses as the reviewer’s comments.

Lines 66-67: ‘The gene coding…E. coli…’ – Repetition from above.

(Response)

We removed the similar paragraph in the Abstract and described more detailed explanation about the strain used in this study in the Materials and Methods as the reviewer’s comments.

Line 68: ‘LB medium’ – acronym used for the first time in text – needs explaining.

(Response)

We revised the sentence as the reviewer’s comments.

Line 74: ‘120 ml of medium in 5000 mL Erlenmeyer’ – Is this correct? Why would you use such a small volume in such a big flask?

(Response)

We found a mistake and changed 5000 mL to 500 mL in the revised manuscript.

Line 82: ‘12000 xg’ – That seems very high. Is it correct?

(Response)

We added the model of the centrifuge used in this study. “g” means the relative centrifugal force not rpm.

Line 84: Replace ‘weight of 100-105oC’ with ‘at’

(Response)

We revised the sentence as the reviewer’s comment.

Line 87: ’20 in’ – Correct with 20 min.

(Response)

We revised the word as the reviewer’s comment.

Line 87: ‘DNS method’ – This method should be briefly explained even if referenced.

(Response)

We added more detailed method of DNS as the reviewer’s comments.

Line 88: ‘calibration curve prepared..’ – Add ‘was prepared’.

No mention of how many replicates were done for these experiments.

The aeration rates and agitation speeds used were described in the Results section instead of Methods.

No mention of impeller type in either the 7L bioreactor or the 100L bioreactor. In fact no mention of what type of bioreactors are.

No mention of feeding regimes or bioreactor operation modes.

(Response)

We revised the line 88 as the reviewer’s comments. We described types of bioreactors and impellers as well as the number of replicates in the Materials and Methods in the revised manuscript as the reviewer’s comments. The production of CMCase was conducted with the batch fermentation. We did not need the feeding materials into bioreactors.

Line 91: Those two equations should be presented independently and not as one.

(Response)

We revised those as the reviewer’s comments.

Line 102: How was the optimal agitation determined and what does optimal mean in this context? It is not clear.

The equations presented in the Results section should be presented in the Methods instead.

(Response)

According to the reference 17, the optimal conditions of the agitation speed and aeration rate of the 7 L bioreactor for the cell growth and the production of CMCase by E. coli KACCP 91335P were determined based on the measured cell growth and produced CMCase from experiments using the response surface methodology. We described these in the revised manuscript.

Line 229: Past tense instead of present tense.

Agitation speeds presented as rpm are not appropriate when comparing bioreactors at different scales. Instead energy dissipation rates should be used.

(Response)

We changed the tense. And we agreed with the reviewer’s comment on comparing bioreactors at different scales. However, comparing in the energy dissipation rates of different scales of bioreactors is not easy. We preferred the shearing forces for comparing bioreactors at different scales.

Figure 5 contains multiple parameters in one graph which makes it confusing.

(Response)

Figure 5 was the comparison of cell growth and the production of CMCase with three different conditions. Each graph in Figure 5 contained the profiles of pH, cell growth (DCW), dissolved oxygen in medium of bioreactors, and production of CMCase, which were very basic parameters. Each graph in Figure 6 showed the cell growths and production of CMCase with different conditions, which was easier to understand their differences.

Line 263: ‘…for the production of wine’ – It is not clear what the connection to wine production is.

(Response)

We revised the sentence as the reviewer’ comments.

Round 2

Reviewer 1 Report

The authors were receptive to the comments, hence I have no other further comments on the content of the work. English proofreading may be necessary to improve the manuscript, and this may be done by the editorial staff or the authors themselves.

Author Response

(Reviewer 1)

Comments and Suggestions for Authors

The authors were receptive to the comments, hence I have no other further comments on the content of the work. English proofreading may be necessary to improve the manuscript, and this may be done by the editorial staff or the authors themselves.

(Response)

Thank for the reviewer’s very valuable comments to improve our manuscript. We revised some parts of the newly revised manuscript regarding to other reviewers’ comments. We marked the revised parts with the red color.

Reviewer 2 Report

     This article describes the effects of shift in agitation speed and aeration rate on the production of carboxymethylcellulase. I feel that the shift of the two factors was effective, and the enhancement of enzyme productivity was significant if the enzyme has enough value to be produced at large scale. Although information in this article is of interest, there are some points which should be corrected or revised as shown below.

Lines 3 and 20: “Escherichia coli KACC 91355P” This should be corrected to “the recombinant Escherichia coli strain” because it has been noticed that wild type E. coli strain has little cellulase activity. In the case of line20, it should be corrected to “the recombinant Escherichia coli KACC 91335P expressing CMCase gene of Bacillus velezensis A-68”.

Lines 17-18: The first sentence is not suitable for the abstract section because it is unclear whether the term cellulase in this sentence means CMCase or not, and this sentence probably describe the previous result. So, this sentence should be deleted.

Line 58-60: The property of Escherichia coli KACC 91335P has not been described, in spite that this strain appears for the first time in the text. I think that the removed sentence of the last manuscript “ The gene coding ….. 91335 P [16].” should be placed before the sentence of present manuscript “The production of CMCase …. growth to the production of CMCase [15].”

Lines 159-165: The statement about the growth phase of microorganisms should be moved to the former section 3.1.

The origin of CMCase gene is B. velezensis, but there is no comparative data about CMCase productivity between the wild type B. velezensis and the recombinant E. coli. Please show the data.

Although the same enzyme is produced, the pattern of production is different between B. velezensis and the recombinant E. coli. I think there are some reasons, and please discuss about that.

Minor comment

Line 48: “overcome in”   “in” should be removed.

Line 58: E.coli is duplicated.

Line 180: “0.05” should be corrected to “0.5”.

Author Response

(Reviewer 2)

Comments and Suggestions for Authors

This article describes the effects of shift in agitation speed and aeration rate on the production of carboxymethylcellulase. I feel that the shift of the two factors was effective, and the enhancement of enzyme productivity was significant if the enzyme has enough value to be produced at large scale. Although information in this article is of interest, there are some points which should be corrected or revised as shown below.

(Response)

Thank for the reviewer’s very valuable comments to improve our manuscript. We revised some parts of the newly revised manuscript regarding to other reviewers’ comments. We marked the revised parts with the red color.

Lines 3 and 20: “Escherichia coli KACC 91355P” This should be corrected to “the recombinant Escherichia coli strain” because it has been noticed that wild type E. coli strain has little cellulase activity. In the case of line20, it should be corrected to “the recombinant Escherichia coli KACC 91335P expressing CMCase gene of Bacillus velezensis A-68”.

(Response)

We revised those in the revised manuscript as the reviewer’s comments.

Lines 17-18: The first sentence is not suitable for the abstract section because it is unclear whether the term cellulase in this sentence means CMCase or not, and this sentence probably describe the previous result. So, this sentence should be deleted.

(Response)

We agreed with the reviewer’s comments and deleted this sentence in the revised manuscript.

Line 58-60: The property of Escherichia coli KACC 91335P has not been described, in spite that this strain appears for the first time in the text. I think that the removed sentence of the last manuscript “ The gene coding ….. 91335 P [16].” should be placed before the sentence of present manuscript “The production of CMCase …. growth to the production of CMCase [15].”

(Response)

We found a mistake and corrected the name of strain from E. coli KACC 913345P to E. coli KACC 913344P in the line of 57 in the revised manuscript. The former name of E. coli KACC 913344P was E. coli JM109/A-53, which is different from E. coli KACC 913345P (formerly E. coli JM109/A-68) used in this study. We wanted to describe that shits in optimal conditions from the cell growth to the production of cellulase enhanced the productivity along with references including the reference 15, which described on E. coli KACC 913344P (formerly E. coli JM109/A-53).

Lines 159-165: The statement about the growth phase of microorganisms should be moved to the former section 3.1.

(Response)

We agreed with the reviewer’s comment. We move those sentences into the section 3.1 in the revised manuscript.

The origin of CMCase gene is B. velezensis, but there is no comparative data about CMCase productivity between the wild type B. velezensis and the recombinant E. coli. Please show the data.

(Response)

According to the reference 16 in the revised manuscript, the production of CMCase by B. velezensis A-68 was 83.8 U/mL, whereas that of E. coli KACC 913345P (formerly E. coli JM109/A-68) was 880.2 U/mL. The production of CMCase by the recombinant was 10.5 times higher than that of the wild strain. In this study, the production of CMCase by E. coli KACC 913345P with shits in optimal conditions of the aeration rate and agitation speed on the pilot scale was 1,212 U/mL, which was 1.36 times higher than that without shifts. We described these in the revised manuscript.

Although the same enzyme is produced, the pattern of production is different between B. velezensis and the recombinant E. coli. I think there are some reasons, and please discuss about that.

(Response)

The CMCase produced by B. velezensis A-68 seemed to be essential for its survival, which could be the growth associated-product. However, CMCase in the recombinant strain of E. coli with might not be essential for its cell growth. Moreover, the wild strain is Bacillus species, whereas the recombinant is E. coil strain. As described in the Introduction, the production patterns by most recombinant strains of E. coli showed the mixed-growth-associated. We described these in the revised manuscript.

Minor comment

Line 48: “overcome in”   “in” should be removed.

(Response)

We removed “in” in the revised manuscript as the reviewer’s comment.

Line 58: E.coli is duplicated.

(Response)

We removed “E. coli” in the revised manuscript as the reviewer’s comment.

Line 180: “0.05” should be corrected to “0.5”.

(Response)

We corrected the number in the revised manuscript as the reviewer’s comment

Reviewer 3 Report

Overall, the manuscript has been improved, but still requires more improvement. The abstract was amended and it now has a good structure, however it is difficult to read and follow and it is confusing in places. It could be better written. The introduction has been minimally amended and more background is still needed in order to understand the overall purpose of this study and the wider impact. The methodology is still missing critical detail like the specific primers used which makes the reproducibility of this study limited. The discussion is still lacking justifications. Minimal evidence of critical judgement.

Specific comments:

Line 28: ‘the simple and economic process’ – This statement needs justification why it is simple and why it is economic. => The justification provided still doesn’t explain the claims of economic.

Line 73: What are those specific primers? Again no details are provided to allow replication of work.

Line 82: ‘12000 xg’ – That seems very high. Is it correct? => This was not answered. Is this value correct?

Lines 257-261: Agitation speeds presented as rpm are not appropriate when comparing bioreactors at different scales. Instead energy dissipation rates should be used. => How do you define shear rate? The authors answer is not correct. Energy dissipation rates are more appropriate for comparing agitation in different scales. See Alvin Nienow’s work. Also no actual values were given.

Figure 5 contains multiple parameters in one graph which makes it confusing. => This graph was not amended. It could be better presented for clarity as separate graphs with maximum 2 parameters per graph and not 4 parameters in one, especially given that every parameter comes with different scales.

Author Response

(Reviewer 3)

Comments and Suggestions for Authors

Overall, the manuscript has been improved, but still requires more improvement. The abstract was amended and it now has a good structure, however it is difficult to read and follow and it is confusing in places. It could be better written. The introduction has been minimally amended and more background is still needed in order to understand the overall purpose of this study and the wider impact. The methodology is still missing critical detail like the specific primers used which makes the reproducibility of this study limited. The discussion is still lacking justifications. Minimal evidence of critical judgement.

(Response)

Thank for the reviewer’s very valuable comments to improve our manuscript. We revised some parts of the Abstract, Material and Methods, and Results and Discussing in the revised manuscript based on all reviewers’ comments. We also revised some parts of the newly revised manuscript regarding to other reviewers’ comments. We marked the revised parts with the red color.

Specific comments:

Line 28: ‘the simple and economic process’ – This statement needs justification why it is simple and why it is economic. => The justification provided still doesn’t explain the claims of economic.

(Response)

We changed the word ‘economic’ to ‘economical’ in the revised manuscript. As described in the late part of the Results and Discussion in the revised manuscript, the enhanced production of CMCase by E. coli KACC 91335P with the conventional multistage process needs at least two bioreactors. One is for the cell growth and the other is for the production of CMCase. However, the cost for the set-up and maintenance of this system is higher than a batch system. Transferring cell culture from the bioreactor to the other bioreactor has some risks involving contamination. The significance and originality of this study is optimizing the time for shifts in optimal conditions from cell growth to production of CMCase in a bioreactor. The process developed in this study needs only a bioreactor for the enhanced production of CMCase by E. coli KACC 91335P. And this process is easy to control the aeration rate and agitation speed in a bioreactor and does not even need to transfer cells from a bioreactor to the other bioreactor.

Line 73: What are those specific primers? Again no details are provided to allow replication of work.

(Response)

We added the DNA sequences of two specific primers in the revised manuscript.

Line 82: ‘12000 x g’ – That seems very high. Is it correct? => This was not answered. Is this value correct?

(Response)

As described in the previous response, the “g” means the relative centrifugal force not rpm. In this revised manuscript, we used rpm of the centrifuge used in this experiment with the model and manufacture.

Lines 257-261: Agitation speeds presented as rpm are not appropriate when comparing bioreactors at different scales. Instead energy dissipation rates should be used. => How do you define shear rate? The authors answer is not correct. Energy dissipation rates are more appropriate for comparing agitation in different scales. See Alvin Nienow’s work. Also no actual values were given.

(Response)

We searched and tried to understand some article of Alvin Nienow’ works as the reviewer’s recommendation. We could not fully understand those articles on the large-scale bioprocess development with our knowledge based on the microbial production on the small-scale. We realized that the impeller tip speed was an essential parameter when the agitating liquid contained solid particles. We calculated the agitation speed of a 100 L bioreactor based on diameters of 7 and 100 L bioreactor. The unit of the impeller tip speed used in the revised manuscript was ‘cm/s.’ We described these with a new reference (No. 27) in the revised manuscript. Than you again for improving our knowledge.

Figure 5 contains multiple parameters in one graph which makes it confusing. => This graph was not amended. It could be better presented for clarity as separate graphs with maximum 2 parameters per graph and not 4 parameters in one, especially given that every parameter comes with different scales.

(Response)

We agreed with the reviewer’s comments. The separation of graphs with 2 parameters should be better for clarity. However, there should be also the space limitation. We thought that the cell growth of E. coli KACC9133P and its production of CMCase were more important. So, we deleted parameters of pH and dissolved oxygen in graphs. Thank you again for your valuable comments for improving our manuscripts.
